# A New Method for Sequential Fractionation of Nitrogen in Drained Organic (Peat) Soils

**DOI:** 10.3390/ijerph20032367

**Published:** 2023-01-29

**Authors:** Marcin Becher, Dorota Kalembasa, Stanisław Kalembasa, Barbara Symanowicz, Dawid Jaremko, Adam Matyszczak

**Affiliations:** Faculty of Agrobioengineering and Animal Husbandry, Siedlce University of Natural Sciences and Humanities, B. Prusa 14 St., 08-110 Siedlce, Poland

**Keywords:** soil nitrogen, sequential extraction, acid hydrolysis, peat degradation

## Abstract

The aim of this study was to assess the transformation of organic matter in organic soils undergoing a phase of secondary transformation, based on a new method of nitrogen compound fractionation. Laboratory tests were carried out for 31 layers of muck (after secondary transformation) and peat (parent material of the soil) of drained organic soils (peat). The new method consists of sequential extraction in the following steps: (1) 0.5 M K_2_SO_4_ (extraction at room temperature); (2) 0.25 M H_2_SO_4_ (hot hydrolysis) (3) 3.0 M H_2_SO_4_ (hot hydrolysis); and (4) concentrated H_2_SO_4_ (mineralization of the post-extraction residue). As a result of the extraction process, the following fractions (operating forms) were obtained: mineral nitrogen (Nmin), dissolved organic nitrogen (N-DON), readily hydrolyzing organic nitrogen (N-RH), non-readily hydrolyzing organic nitrogen (N-NRH), and non-hydrolyzing organic nitrogen (N-NH). The study demonstrates the usefulness of the applied method for assessing the degree of secondary transformation of drained organic soils. The obtained results of nitrogen fractionation indicate the significant dynamics of nitrogen forms’ transformations and a significant relationship between these forms and soil properties. Nitrogen transformation processes during the secondary transformation process after dehydration resulted in an increase in the share of N-DON (on average: 1.47% of N_org_ for the peat layers and 2.97% of Norg for the muck layers) and in an increase in the share of N_RHON_ (on average: 20.7% of Norg for the peat layers and 33.5% of Norg for the muck layers). The method of sequential nitrogen fractionation used in our study allowed us to define an index determining the degree of transformation of organic matter in peat after drying. We defined it as the ratio of readily hydrolyzable forms (the fraction is very variable in the secondary transformation process) to non-readily hydrolyzable forms (relatively stable fraction in the secondary transformation process): N-RH/N-NRH. The average value of this index was significantly lower in the peat layers (0.64 on average) than in the muck beds (1.04 on average). The value of this index is significantly correlated with soil properties: bulk density (R^2^ = 0.470); general porosity (R^2^ = 0.503); total carbon content (TC) (R^2^ = 0.425); total carbon to total nitrogen ratio (TC/TN) (R^2^ = 0.619); and share of carbon of humic substances (C-HS) (R^2^ = 0.466). We believe that the method of sequential nitrogen fractionation may be useful for other soils and organic materials.

## 1. Introduction

Peatlands (terrestrial wetland ecosystems) are responsible for the accumulation of about 30% of the Earth’s total surface organic carbon resources and therefore, they are one of the most important repositories of organic matter, despite their relatively small area of about 3% of land area [1,2]. The function of carbon sequestration by peatlands in the industrial era was significantly disturbed, caused by climate change and direct human activity, mainly drainage [3,4].

Drainage of peatlands for agricultural purposes is the most important cause of the degradation of temperate low peatlands [5]. In addition, the climate change forecasts for Central Europe, which predict a decrease in annual precipitation and an increase in air temperature, are worrying [6,7]. Such climatic changes will increase the rate of soil evapotranspiration and decrease the depth of the water level [8,9]. Peatlands in Poland cover approximately 1.2 million ha, which is around 4% of the country’s area. About 90% of the peatlands outside of forests in Poland have been dehydrated due to drainage conducted to include hydrogenic habitats in areas under agricultural production and because of river training and industrial investments [10,11]. The condition of peatland ecosystems and the evolution of occurring soils depends to a large extent on water [12,13,14,15]. Dehydration breaks organic matter accumulation and changes the direction of soil evolution by making it enter the decession phase (organic matter loss). Progressing aeration of surface soil layers alternating with periodical rehydration intensifies the physical, biological, and chemical processes of the parental soil material. This phenomenon is called soil secondary transformation. The scope of changes in organic soil morphology and of modification of soil material properties affected by this process is the basis of distinguishing murshic organic soils [16,17].

It is common knowledge that organic soils are a very important reservoir of surface biogenic elements, including nitrogen. Due to the scale of organic matter accumulation and the dynamics of its changes, they are a link in the nitrogen biogeochemical cycle which is of paramount importance [3,18,19,20]. Nitrogen is an element characterized by the marked dynamics of variability in the forms occurring in soils which include, e.g., biological sorption, mineralization, ammonification, and nitrification [21,22,23]. Nitrogen accumulation in organic soils depends on the peat type and is usually reversely proportional to the depth, and the nitrogen content of organic soil may increase with its age and development [1,16,18,24,25].

One of the most important results of secondary transformation is organic matter mineralization, combined with gas emission and nutrient release—possible results of these transformations, such as water eutrophication and greenhouse effects, have become global problems, going beyond the environment of the very soil [26,27,28,29,30,31,32].

The processes of release and migration of biogenic elements during excessive mineralization of organic (peat)matter are one of the causes of eutrophication of surface waters. This phenomenon is extremely important in the case of soils covering wetland areas, which constitute a border zone between the hydrosphere and the lithosphere [33,34]. The most important element responsible for eutrophication is nitrogen, as it regulates biological processes and limits biomass production [18,22,29].

Scientific studies have demonstrated that secondary transformation determines far-reaching quantitative and qualitative changes in organic matter composition [1,35]. Intense mineralization is accompanied by humification—secondary humification under certain conditions—which leads to an increase in the soil content of humic substances [36,37,38]. It has been demonstrated that secondary transformation transforms peat so that there is an increase in the potential nitrogen availability of the peat or, in other words, there is an increase in the amount of nitrogen in organic compounds prone to mineralization [1,16,24,37,39].

Qualitative studies of organic matter in organic soils are usually based on the share of carbon in separated fractions (fractionation of organic matter). This applies to humic substances, where an alkaline extraction process is used to separate them, e.g., [38,40,41,42]. Similarly, in the case of hot acid hydrolysis, research has been conducted mainly in the context of soil carbon sequestration and estimation of the stock of labile and stable carbon in the environment, e.g., [43,44,45,46,47,48,49,50]. In the case of soil nitrogen, the hot acid hydrolysis process is usually used in a single-stage version, e.g., for the separation and qualitative testing of nitrogen-containing organic compounds, e.g., [51,52,53].

In our studies on organic soils, we propose a new method of sequential fractionation of nitrogen compounds. In the first stage, the method consists of extraction with a neutral reagent (0.5 M K_2_SO_4_), and then in two-phase hot acid hydrolysis, with different concentrations of H_2_SO_4_ (0.25 M and 3.0 M). Two-step acid hydrolysis using different concentrations of H_2_SO_4_ has been used in many scientific studies to determine the stock of organic matter susceptible and resistant to decomposition, e.g., [54,55,56,57]. Different concentrations of mineral acids and different extraction times were used, and dissolved organic matter (DOM) was usually not taken into account.

So far, scientific research using acid hydrolysis techniques has mainly concerned mineral soils and the decomposition of plant remains. We claim that there is a need to extend scientific research on chemical speciation using the technique of acid hydrolysis in organic peat soils of the temperate climatic zone. First of all, because they are largely subject to anthropogenic pressure, e.g., dehydration and intensive agricultural use. In addition, they react relatively quickly to climate warming by accelerated mineralization of organic matter and the dispersion of elements in the environment. The problem of renaturation of drained peatlands is very important and current [5,34]. Therefore, we believe that the proposed method of sequential nitrogen fractionation may contribute to better management of these soils, including more rational management of nitrogen in these soils.

The main research problem was to check whether the information obtained as a result of sequential fractionation of nitrogen compounds can be used as a basis for the assessment of organic matter transformations in drained peat soils. The previous research indicates that possibility [40,58]. Therefore, we make the following hypotheses:–Sequential fractionation of nitrogen compounds can be useful to assess the degree of organic matter transformation in drained organic (peat) soils.–The share of labile fractions of nitrogen in relation to the stable fractions of this element can be the basis for determining chemical indices assessing the degree of secondary transformations of organic matter after drainage of the peatlands.

## 2. Materials and Methods

### 2.1. Study Area and Material

Field soil research was carried out on selected fen peatlands in central-eastern Poland in the catchment area of the Bug River. In the 1950s and 1960s, the study area was drained. Since then, the organic soils of these peatlands have been intensively used for agriculture and are under strong pressure due to drainage and secondary transformations [10,40]. According to the Köppen–Geiger climate classification [59], the examined peat bogs are located in a fully humid, warm temperate climate zone with warm summers.

Eight profiles of organic soils were selected for the study which, according to the Polish Soil Classification [17], were classified as follows (the English names of soil types are given according to Świtoniak et al., 2016 [60]:–*Hemi-murshic soils* (five profiles; 52°06′38.2″ N, 22°37′05.5″ E; 52°06′57.7″ N, 22°36′39.6″ E; 52°08′35.6″ N, 22°34′06.0″ E; 52°08′47.6″ N, 22°33′54.8″ E; 52°03′30.6″ N, 22°19′32.0″ E), formed from reed and sedge peats (moderately decomposed, H3–H6 on the von Post scale) [61];–*Sapri-murshic soils* (three profiles; 52°10′42.1″ N, 22°28′57.8″ E; 52°10′46.5″ N 22°29′18.3″ E, 52°03′32.5″ N, 22°19′49.8″ E), formed from alder peats (heavily decomposed, H7–H8 on the von Post scale).

In the FAO WRB Soil Classification [62], the soils selected for testing were defined as: *Eutric Murshic Hemic Histosol* and *Eutric Murshic Sapric Histosol*.

In the vegetation period, the groundwater level was 60–75 cm below the ground level. Morphological characteristics allowed to us to distinguish 31 soil layers [17], including:–Sixteen layers with a clearly developed grainy structure meeting the criteria of the murshic layer, typical for the upper parts of drained organic soils. The thickness of these layers ranged from 28 to 35 cm.–Fifteen layers meeting the criteria of the histic layer occurring under the *murshic* layers (9—rush and sedge peats and 6—alder peats). The samples were taken below the muck layers to a depth of 80 cm.

A representative soil sample was taken from each selected soil layer (*n* = 31). All of the layers were subjected to detailed laboratory analyses. The laboratory analyses were performed in triplicate.

### 2.2. Laboratory Analyses

#### 2.2.1. Soil Properties

During the field studies, genetic horizons were identified, and samples were taken for testing. Soil cores were placed in cylinders (v = 100 cm^3^, in triplicate) to determine physical soil properties (bulk density and total porosity) by methods that are common in pedological studies [63].

The soil samples were dried at 40 °C, crushed manifold during drying, and, when dried, placed on a sieve (ø 2.0 mm). Part of the dried sample was ground in an agate pestle and mortar (to ø < 0.25 mm) and analysed chemically (in triplicate). The results of the analyses were calculated on the base of absolute dry mass which was determined after drying the soil samples at 105 °C. The following soil properties were determined:–pH in 1M KCl, soil/solution = 1/5 (*v*/*v*);–Ash content, after combustion at 550 °C, for 6 h;–Total carbon (TC) content and total nitrogen (TN) content, determined by an elemental analyzer Series II 2400, produced by Perkin Elmer (Waltham, MA, USA);–Humic substances separated by extraction with 1 M NaOH (m/v = 1/50, extraction time 24 h). Carbon and nitrogen contents were determined in the extract (after centrifugation, 4000 rpm). The carbon and nitrogen of humic substances were obtained (C-HS and N-HS ).

#### 2.2.2. Sequential Nitrogen Fractionation

Sequential fractionation was conducted which consisted of extraction with a neutral reagent and two-phase acid hydrolysis. The following solutions were used (2 g soil samples, m/v = 1:25) [23,40]:–0.5 M K_2_SO_4_ to extract nitrogen in mineral compounds and readily soluble organic compounds. The extraction was conducted at room temperature for 24 h;–0.25 M H_2_SO_4_ to extract organic compounds which are operationally called “readily hydrolyzable compounds”. Hydrolysis was carried out for 4 h at the boiling temperature of the mixture, under a water reflux condenser;–3.0 M H_2_SO_4_ to extract organic compounds which are operationally called “non-readily hydrolyzable compounds” (see above).

Solution clarity was obtained by centrifuging (4000 rpm) and filtering to a flask through cellulose filter paper (v = 100 mL). After each extraction/hydrolysis stage, the soil samples were rinsed with deionized water three times (15 mL) and the proper extract was refilled. Soil particles remaining on the filter were rinsed with deionized water into extraction flasks and evaporated until dry.

The method of sequential fractionation applied made it possible to define different forms (operational fractions) of soil nitrogen (Table 1).

#### 2.2.3. Determination of Carbon and Nitrogen in Solutions

The solutions for elemental determination were filtered (ø 0.45 μm) using negative pressure. The carbon content was determined by an oxidation and titration method [64]. The nitrogen content in the solutions was determined by the Kjeldahl method [65]. Mineralizationtion of nitrogen organic compounds was performed in Kjeldahl flasks from which ammonium was distilled immediately after mineralization. Omission of sample dilution and the use of a micropipette for precipitation made it possible to determine the nitrogen in samples where the nitrogen concentration was low. The analytical procedure of mineralization and determination of nitrogen organic compounds was checked using acetanilide (Table 2).

Moreover, the aggregate nitrogen amount of separated fractions was compared to the total nitrogen content in the sample (Table 3).

### 2.3. Statistical Calculations

Statistical calculations were performed using STATISTICA 12 PL (TIBCO Software Inc., Palo Alto, CA, USA). The following descriptive statistics were used to characterize the populations: arithmetic mean, minimum value, maximum value, standard deviation, and regression coefficient. Associations between the characteristics studied were expressed by means of a simple correlation coefficient (r) and, in selected cases, linear regression equations were calculated. In order to show significant differences between the examined parameters of soil genetic levels, one-factor analysis of variance and a post-hoc Tukey’s test were carried out.

## 3. Results and Discussion

### 3.1. Soil Properties

Changes in morphological characteristics during the secondary transformation in the *soils* studied were confirmed by the results of laboratory analyses as compared with the peat layers. Statistical analysis showed that the murshic and histic layers differ significantly in terms of most of the tested soil properties. The murshic layers of these soils had higher ash contents, greater soil mass density, and a lower share of soil pores (Table 4). Moreover, it was demonstrated that the murshic layers had lower carbon contents and a narrower TC/TN ratio compared with the histic layers (Table 5).

The nitrogen content was characterized by a lower profile variability. The murshic and histic layers constituted homogeneous groups. This suggests that in our research, the nitrogen content in the murshic layers remained at a level close to the “starting” value (i.e., before the protective effect of water on the organic matter was removed). This confirms the research of other authors, which shows that secondary nitrogen accumulation (increased concentration) may even occur in the murshic layers [20,24,34,40].

At the background of carbon and nitrogen transformations, a narrow TC/TN ratio was found in the murshic layers compared with the indigenous peat (histic layers). The TC/TN ratio obtained (10.7–18.4), accompanied by low acidity (pH_KCl_ 5.22–6.36), is indicative of the entropy of the soil environment examined as well as high biological activity and substantial organic matter processing as a result of mineralization and humification processes [36,48,66,67].

The soil properties and the results of secondary transformations found in this study are typical for dehydrated peat, in which the carbon content decreases in the drained and warmer surface layers as a result of aerobic processes of mineralization of organic compounds and the release of carbon dioxide CO_2_ into the atmosphere [4,68,69]. Although, the increased primary production of the current bog vegetation formation may partly compensate for this phenomenon [70]. In addition, higher groundwater levels can weaken microbial activity and thus slow down or even stop the process of mineralization of organic matter [71].

Carbon and nitrogen extracted from the soil samples using an alkaline solution (0.1 M NaOH) mainly represented the pool of these elements bound by humic substances [38,40,41,42]. In the study, a slightly lower extraction efficiency of nitrogen compounds was achieved compared with carbon. In general, the carbon and nitrogen share of humic substances was characterized by marked variability associated with the advancement of the organic matter humification process. Thus, the significantly higher share of these elements in the 0.1 M NaOH extract was found in the murshic layers whereas, for the peat layers, a higher share was determined in osier peats (*sapric* material) compared with reed and sedge peats (*hemic* material). By analyzing the content of carbon and nitrogen in the alkaline extract, we first of all proved the differences between the muck and peat layers regarding the chemical composition, especially the content of humic substances. Thus, our research confirmed that the share of humic substances in peat is positively correlated with the degree of peat decomposition [25,38,69].

### 3.2. Correlation of Soil Properties

The physical properties and carbon content of the soils studied were closely correlated with the degree of organic matter decomposition, including the secondary transformation advancement, which is confirmed by significant correlation coefficients calculated for the soil properties studied (Table 6) where the degree of soil organic matter transformation is expressed by such parameters as a quantitative carbon to nitrogen ratio (TC/TN) and the share of these elements in humified organic matter (C-HS and N-HS).

The dynamics of changes in the TC/TN value were significantly affected by the variability of carbon content (positive correlation), with the effect of nitrogen accumulation being insignificant. As expected, a significant relationship was found which indicated that the TC/TN value in the soil mass declined as the degree of organic matter decomposition increased (negative correlation with C-HS and N-HS).

In general, the data obtained confirm the notion dominating in the literature about the usefulness of the quantitative carbon-to-nitrogen rate and the share of humic substances for the description of organic matter transformation advancement occurring in dehydrated organic soils [16,20,40,72,73].

### 3.3. Nitrogen Fractions after Extraction with a Neutral Reagent and Acid Hydrolysis

The obtained results of the fractionation of mineral and organic nitrogen compounds revealed marked dynamics of various nitrogen forms and their close association with soil properties (Table 7 and Table 8).

Highly varied amounts of mineral nitrogen forms (N-NH_4_ + N-NO_x_) were identified in the soil samples. A slightly higher share of these forms was determined in the murshic layers, which was probably due to an inflow and intensive mineralization of fresh organic matter from the vegetation cover as well as other sources such as fertilization and the reduction of elemental nitrogen by microorganisms; it is also well known that this nitrogen form undergoes marked seasonal dynamics [16,24,74,75]. In the examined soil population, the share of mineral nitrogen forms was also positively influenced by declining acidification (r = 0.46).

In the genetic horizons of the soils studied, organic nitrogen constituted from 96.9 to 99.6% of the total nitrogen content. The fractionation results are presented as the share of nitrogen fraction in Norg.

An application in the sequential analysis of reagents characterized by increasing ‘extraction power’ (0.5 M K_2_SO_4_ < 0.25 M H_2_SO_4_ < 3.0 M H_2_SO_4_) made it possible to identify soil organic nitrogen in various forms (arbitrary operational forms): soluble (N-DON), readily hydrolyzable (N-RH), non-readily hydrolyzable (N-NRH), and non-hydrolyzable (N-NH). The identified nitrogen forms represent organic compounds with different potential stabilities in the soil environment: N-DON < N-RH < N-NRH < N-NH. The order reflects the reverse relationship of readiness for decomposition processes.

The share of soluble organic nitrogen (N-DON) clearly had the highest variability and the lowest share compared with the remaining organic forms. The share of this fraction was closely associated with the processes of organic matter decomposition (including secondary transformation), as indicated by significant differences between the murshic and histic layers and by significant values of correlation coefficients with soil properties. Ready solubility and potential mobility of organic compounds extracted using 0.5 M K_2_SO_4_ probably added to the variability in the share of this fraction in the soil profiles studied [27].

In general, organic compounds extractable with a solution of neutral salt represent the so-called soluble part of soil organic matter (DON) which may be composed of numerous groups of compounds containing nitrogen such as amino acids, amino sugars, proteins, low-molecular-weight fractions of humic acids and many other simple organic compounds. Although the share of this organic matter fraction and its stability are low, the fraction is ascribed the highest reactivity in soil processes [26,66,76,77,78,79].

In the method of sequential fractionation, the sum of nitrogen of organic compounds extracted from soil samples by means of two-phase acid hydrolysis, with different concentrations of hydrogen ions (0.5 and 6.0 mol dm^−3^), was operationally defined to be hydrolyzable organic nitrogen (N-H). The share of this nitrogen in the soils studied ranged from 45.8 to 73.8% of organic nitrogen stores and was clearly lower in the histic layers than in the murshic layers. The hydrolyzable fraction consists mainly of proteins, nucleic acids, and polysaccharides [54,57,80].

In general, examination of nitrogen compounds that make up soil organic matter includes their hydrolysis with solutions of mineral acids or bases as well as qualitative analysis of the hydrolysates obtained [22,51,52,53,54,80,81]. Often used in the hydrolysis technique is 6.0 M hydrochloric acid, which separates labile (hydrolyzed) compounds from resistant (non-hydrolyzed) compounds [45,46,48]. It is believed that hydrolysis with mineral acids simulates the stability of SOM against hydrolytic decomposition caused by extracellular enzymes of soil microorganisms [47]. In our research, we used sulfuric acid due to the specific nature of peat organic matter, which is largely composed of plant tissues. H_2_SO_4_ is more effective than HCl in the hydrolysis of organic matter, especially plant tissues [81]. According to Rovirai Vallejo [57], the method of hydrolysis with H_2_SO_4_ is influenced by the mineral soil matrix in relation to HCl, which is of very little importance in our research, because we tested organic soil samples. In the study presented here, the second phase of hydrolysis included the application of 3.0 M sulphuric acid (VI) which introduces into the hydrolyzing system the concentration of hydrogen ions corresponding to 6.0 M HCl.

Of the hydrolyzable nitrogen forms, a higher variability of their shares was found for the fraction extracted using 0.25 M H_2_SO_4_, which is operationally called ‘readily hydrolyzable’ (N-RH). The examined genetic horizons may be ranked in the following increasing order of the share of this fraction in Norg: peat (*hemic*) < peat (*sapric*) < murshic layers. Coefficients of correlation with soil properties indicate that accumulation of this nitrogen fraction is significantly caused by an aerobic transformation of organic matter in murshic soil and processes of peat formation decomposition—significant positive correlations (high coefficient values) with organic soil properties which varied in the way which is typical of secondary transformation and organic matter decomposition, in particular a positive relationship with C-HS and N-HS and a negative relationship with the TC/TN ratio. Selected relationships are presented together graphically with the non-readily hydrolyzable fraction which had lower variability and no close associations with soil properties (excluding a negative correlation with TN) (Figure 1).

The share of nitrogen that remained in the soil material after extraction with a neutral reagent and two-phase acid hydrolysis was classified as non-hydrolyzable organic nitrogen (N-NH). In general, unlike the efficiency of extraction in the organic soils studied, a higher share of this nitrogen form is noted in the histic layers than in the murshic layers. Values of correlation coefficients calculated for this fraction indicate that organic matter transformation in the soils studied contributed to a decline in the share of this nitrogen form.

Research has indicated that non-hydrolyzable soil nitrogen fractions may be dominated by derivatives of aromatic and heterocyclic compounds, peptide structures resistant to microbiological decomposition, and lignin and related compounds [22,54,82], as well as fats, waxes, resins, and suberins [54].

The results obtained indicate a substantial variation in the share of individual nitrogen forms during the transformation of the solid phase of the soils studied (secondary transformation) occurring in the direction of an increase in the role of the share of hydrolyzable forms, in particular, readily hydrolyzable and soluble forms. This finding confirms the literature reports claiming that the nitrogen amount in readily mineralizable organic compounds increases during the process of peat decomposition [16,24,39,40].

Based on the obtained results and the relationship between the fractions defined on the basis of the presented sequential nitrogen extraction, the nitrogen index of acid hydrolysis (N-RH/N-NRH) can be proposed. The value of this index is the ratio of the quantitatively significant, variable, easily hydrolyzable form of nitrogen to the quantitatively significant and relatively stable, hardly hydrolyzable form of nitrogen. In our research, the value of this index was in the range of 0.46–1.32. The murshic layers were characterized by a higher value of this ratio—the average values of this ratio allow the tested soil horizons to be classified in the following order: 0.62 (histic layers of hemi-murshic soils), 0.67 (histic layers of sapri-murshic soils), and 1.04 (murshic layers).

As indicated by the correlation coefficients, the value of this index was not affected by the content of total nitrogen in the tested soils, but it was significantly correlated with soil properties changing as a result of the transformation of organic matter under the influence of drainage (Table 8, Figure 1 and Figure 2).

The value of this index significantly increased with an increase in soil density and crude ash content (r = 0.69 and r = 0.56, respectively), with a simultaneous decrease in the value of soil total porosity (r = −0.71), organic matter content (r = −0.65 for TC), and TC/TN (r = −0.79). In addition, it was found that the increase in this index is significantly affected by the transformation of organic matter in the humification process, which in our research is represented by the share of carbon and nitrogen extracted with 0.1M NaOH (r = 0.68 for C-HS and 0.51 for N-HS).

In our research, the ratio N-RH/N-NRH allowed us to assess the advancement of secondary transformations of organic soils after drainage. The application of the method of sequential nitrogen fractionation and the index determined on the basis of hydrolyzing nitrogen will enable the assessment of the process of secondary transformations in organic soils (peat) of a temperate climate, drained for the purposes of agricultural activity. Other studies also indicate such a possibility [39,40,69]. It will probably be possible to apply this method and index to a wider spectrum of organic soils undergoing changes under the influence of drainage. However, this requires further research.

## 4. Conclusions

The applied sequential fractionation of nitrogen compounds based on extraction with a neutral reagent and two-phase acid hydrolysis provided a lot of information about qualitative changes in the solid phase occurring in drained organic soils. The process of aerobic metabolism of organic matter (typical for secondary transformations) increased the share of the soluble fraction of organic nitrogen (N-DON, extracted with 0.5 M K_2_SO_4_) and the share of the readily hydrolyzable fraction (N-RH, extracted with 0.25 M H_2_SO_4_). We found that the share of N-RH is closely correlated with the physical and chemical properties of soils (it increases with the degree of advancement of secondary transformation). We decided that the appropriate index assessing the progress of decomposition processes of organic matter in organic soils may be the ratio of the share of easily hydrolyzing nitrogen (clearly variable in terms of quantity) to the share of hardly hydrolyzing nitrogen (relatively stable in terms of quantity) (N-RH/N-NRH). Significant correlations of N-RH/N-NRH values with soil properties were found. The murshic layers (after secondary transformation) were characterized by higher values of this index in relation to the histic layers (parent soil material).

In our opinion, this method of chemical fractionation of nitrogen compounds can be applied to other soils in order to enrich the knowledge about the advancement of organic matter decomposition processes and to assess the current and potential threat to the mineralization of organic nitrogen compounds (including migration outside the soil environment).

## Figures and Tables

**Figure 1 ijerph-20-02367-f001:**
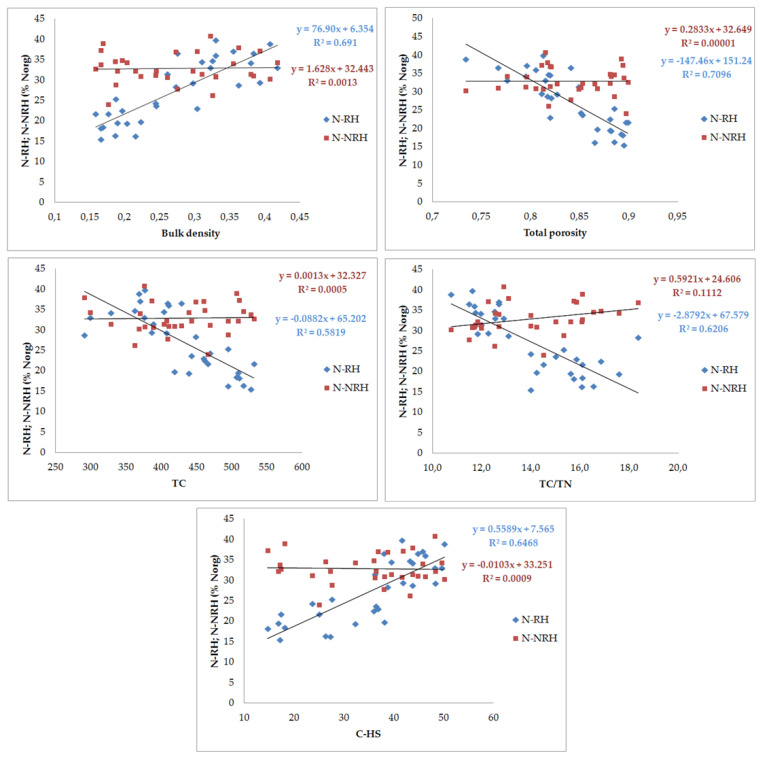
Relationship between percentage of hydrolyzing nitrogen fractions (N_RHON_ and N_NRHON_) and the chosen physical and chemical properties of the soil.

**Figure 2 ijerph-20-02367-f002:**
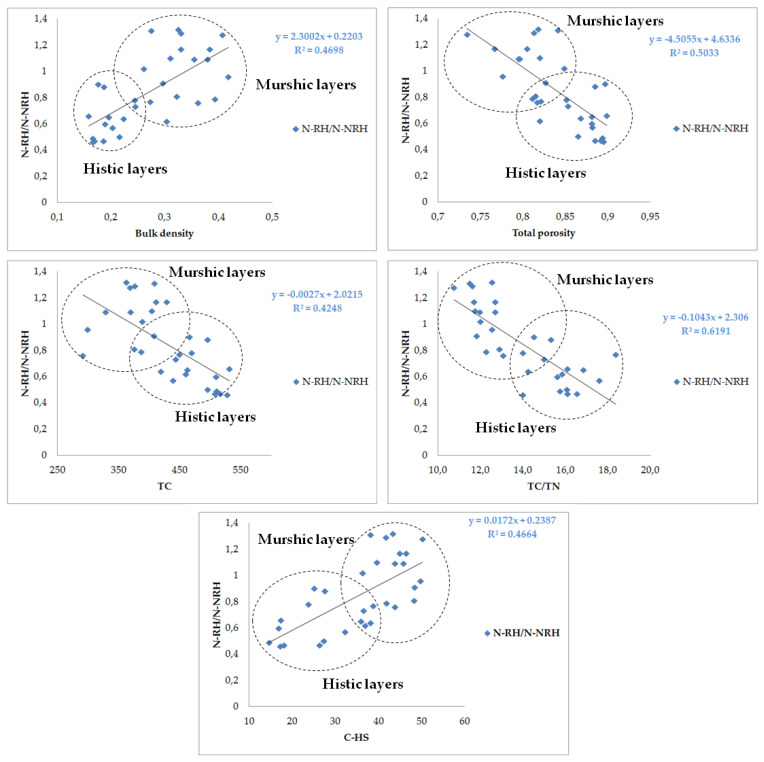
Relationship between the ratio of readily hydrolyzable to non-readily hydrolyzable nitrogen (N_RHON_/N_NRHON_) and the chosen physical and chemical properties of the soil.

**Table 1 ijerph-20-02367-t001:** Nitrogen forms and determination methods.

(Operational) Nitrogen Form	Determination Method
N-NH_4_—ammonium nitrogen	Distillation from 0.5M K_2_SO_4_ extract, after MgO alkalinizationn
N-NO_x_—nitrate nitrogen (III and V)	Distillation from 0.5 M K_2_SO_4_ extract after N-NH_4_ distillation and reduction with a Devard mixture
Nmin—nitrogen in mineral compounds	Nmin = N-NH_4_ + N-NO_x_
Norg—nitrogen in organic compounds	Norg = TN − Nmin
N-K_2_SO_4_	Determined in 0.5 M K_2_SO_4_ extract, after solution mineralization
N-DON—dissolved organic nitrogen	N-DON = N-K_2_SO_4_ – Nmin
N-RH—readily hydrolyzable organic nitrogen	Determined in 0.25 M H_2_SO_4_ hydrolysate, after solution mineralization
N-NRH—non-readily hydrolyzable organic nitrogen	Determined in 3 M H_2_SO_4_ hydrolysate, after solution mineralization (in H_2_SO_4_)
N-H—hydrolyzable organic nitrogen	N-H = N-RH + N-NRH
N-NH—non-hydrolyzable organic nitrogen	Nitrogen determined after mineralization of post-extraction remains in concentrated sulphuric acid (VI)

**Table 2 ijerph-20-02367-t002:** Checking the procedure for the quantitative analysis of nitrogen.

Parameter	Value
Expected value (%)	10.36
Obtained value, *n* = 6 (%) Mean (range) SD	10.32 (10.28–10.40) 0.05

**Table 3 ijerph-20-02367-t003:** Comparison of the aggregated nitrogen content of fractions separated from TN.

Parameter	Value
Recovered (%) = N-K_2_SO_4_ + N-RH + N-NRH + N-NH)/TN × 100 Mean (range) SD	98.01 (95.5–99.7) 1.25

**Table 4 ijerph-20-02367-t004:** Values of descriptive statistics of physical properties and pH.

Parameters	Ash Content (%)	Bulk Density (g cm^−3^)	Total Porosity (m^3^ m^−3^)	pH_KCl_
Mean	22.51	0.273	0.840	-
SD	8.80	0.082	0.043	-
CV (%)	39.13	30.19	5.17	-
Murshic layers (*n* = 16)				
Mean	28.52 (a)	0.341 (a)	0.807 (a)	-
Min	18.40	0.261	0.734	5.22
Max	43.50	0.418	0.849	6.36
Histic layers (*n* = 15)				
Mean	16.09 (b)	0.20 (b)	0.88 (b)	-
Min	10.20	0.16	0.82	-
Max	24.00	0.27	0.90	-
Histic layers of *Hemi-murshic soils* (*n* = 9)				
Mean	14.04	0.19	0.88	-
Min	10.20	0.16	0.85	5.51
Max	24.00	0.24	0.90	6.03
Histic layers of *Sapri-murshic soils* (*n* = 6)				
Mean	20.20	0.23	0.86	-
Min	17.10	0.20	0.82	5.62
Max	23.90	0.27	0.88	6.08

(a), (b)—homogeneous means at α = 0.05.

**Table 5 ijerph-20-02367-t005:** Values of descriptive statistics of carbon and nitrogen contents, the TC/TN rate, and the carbon and nitrogen shares in humic substances.

Parameters	TC (g kg^−1^)	TN (g kg^−1^)	TC/TN	C-HS (% TC)	N-HS (% Norg)
Mean	429.49	30.89	13.98	35.33	38.40
SD	65.82	3.65	2.08	10.95	11.98
CV (%)	15.32	11.81	14.89	31.00	31.21
Murshic layers (*n* = 16)					
Mean	379.5 (a)	30.9 (a)	12.4 (a)	43.7 (a)	46.3 (a)
Min	291.5	22.3	10.7	36.3	29.1
Max	460.3	35.6	15.8	50.2	58.2
Histic layers (*n* = 16)					
Mean	482.9 (b)	30.8 (a)	15.7 (b)	26.4 (b)	30.0 (b)
Min	419.0	24.5	14.0	14.7	17.0
Max	532.0	37.7	18.4	38.8	44.8
Histic layers of *Hemi-murshic soils* (*n* = 9)					
Mean	503.00	32.75	15.40	21.44	26.60
Min	466.32	30.85	13.99	14.70	17.00
Max	531.65	37.71	16.53	27.60	44.80
Histic layers of *Sapri-murshic soils* (*n* = 6)					
Mean	442.60	27.18	16.40	36.36	36.84
Min	419.09	24.48	14.23	32.30	32.50
Max	462.30	29.55	18.35	38.80	43.20

(a), (b)—homogeneous means at α = 0.05.

**Table 6 ijerph-20-02367-t006:** Correlation matrix for physical and chemical soil properties.

Parameter	Ash Content	Bulk Density	Total Porosity	pH_KCl_	TC	TN	TC/TN	C-HS	N-HS
Ash content	-								
Bulk density	0.79	-							
Total porosity	−0.66	−0.96	-						
pH_KCl_	0.31	0.03	0.14	-					
TC	−0.96	−0.87	0.77	−0.24	-				
TN	−0.47	−0,23	0,14	−0.37	0.44	-			
TC/TN	−0.63	−0.71	0.67	0.02	0.69	−0.34	-		
C-HS	0.77	0.88	−0.86	0.02	−0.86	−0.29	−0.63	-	
N-HS	0.57	0.71	−0.70	−0.06	−0.67	−0.08	−0.60	0.76	-

Critical values (*n* = 31): 0.36 (at α = 0.05); 0.46 (at α = 0.01).

**Table 7 ijerph-20-02367-t007:** Values of descriptive statistics of the separated nitrogen fractions.

Parameters	Nmin (%TN)	Norg (g kg^−1^)	N-DON	Hydrolyzable N			Non-Hydrolyzable N N-NH	N-RH/N-NRH
				N-H	N-RH	N-NRH		
			(% Norg)					
Mean	1.14	36.0	2.25	60.20	27.31	32.89	37.56	0.85
SD	0.62	21.5	0.97	7.63	7.61	3.70	8.29	0.28
CV (%)	54.55	59.79	43.13	12.67	27.87	11.24	22.06	32.60
Murshic layers (*n* = 16)								
Mean	1.25 (a)	39.7 (a)	2.97 (a)	66.27 (a)	33.50 (a)	32.77 (a)	30.76 (a)	1.04 (a)
Min	0.64	14.4	1.91	60.00	23.00	26.20	22.70	0.62
Max	3.09	101.0	3.80	73.80	39.80	40.80	36.30	1.32
Histic layers (*n* = 15)								
Mean	1.02 (a)	30.7 (a)	1.47 (b)	53.73 (b)	20.71 (b)	33.01 (a)	44.81 (b)	0.64 (b)
Min	0.41	24.4	0.89	45.80	15.40	24.10	33.30	0.46
Max	2.14	37.5	2.81	65.20	28.30	39.00	53.30	0.90
Histic layers of *Hemi-murshic* soils (*n* = 9)								
Mean	1.13	37.0	1.32	52.31	19.71	32.60	46.40	0.62
Min	0.41	13.6	0.89	45.80	15.40	24.10	40.70	0.46
Max	2.14	69.7	1.89	57.40	25.40	39.00	53.30	0.90
Histic layers of *Sapri-murshic* soils (*n* = 6)								
Mean	0.81	22.2	1.80	56.56	22.72	33.84	41.64	0.67
Min	0.68	17.0	1.48	50.60	19.40	30.90	33.30	0.57
Max	0.89	25.8	2.81	65.20	28.30	36.90	47.90	0.77

(a), (b)—homogeneous means at α = 0.05

**Table 8 ijerph-20-02367-t008:** Correlation of the share of nitrogen fraction with soil properties.

Parameters	Nmin	N-DON	N-H	N-RH	N-NRH	N-NH	N-RH/N-NRH
**Ash content**	0.07	0.70	0.62	0.65	−0.05	−0.66	0.56
**Bulk density**	0.06	0.66	0.85	0.83	0.04	−0.86	0.69
**Total porosity**	0.01	−0.56	−0.84	−0.84	0.00	0.84	−0.71
**pH_KCl_**	0.46	0.34	0.09	0.03	0.12	−0.12	0.00
**TC**	−0.01	−0.76	−0.75	−0.76	0.02	0.78	−0.65
**TN**	0.28	−0.39	−0.16	0.03	−0.39	0.20	0.17
**TC/TN**	−0.23	−0.48	−0.62	−0.79	0.33	0.63	−0.79
**C-HS**	0.03	0.69	0.79	0.80	−0.03	−0.81	0.68
**N-HS**	−0.01	0.53	0.67	0.65	0.06	−0.68	0.51

For critical values, see Table 6.

## Data Availability

Not applicable.

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
