# Peer review of "A New Method for Sequential Fractionation of Nitrogen in Drained Organic (Peat) Soils"

_ijerph, 2023, doi:10.3390/ijerph20032367_

Round 1

Author Response

Dear Reviewer,

Thank you for your review. All your comments have been included in the revised version of the manuscript.

Sincerely

Authors

Reviewer 2 Report

Substantive evaluation of the manuscript.

The manuscript is written in good style, is informative and gives overview of the topic. In this review, I offer a few suggestions as to where certain points can be elaborated upon or revised in the manuscript.

Detailed comments (with reference to lines):

1.     Line 114: Please specify how many samples were taken for analysis

2.     Line 117: Please specify how many samples were taken for analysis

3.     Line 137: Please add a location (city, country) for the company

4.     Table 4, 5, and 7 (page 7 9-10): Complete tables 4, 5 and 7 with the number of analysed samples n=...

5.     Table 6: Correct the "TC/TN" in the table 6

6.     Figure 3 (page 14): Correct the formatting of the figure 3.

7.     The authors did not provide the publication's DOI numbers or the Uniform Resource Locator (URL) in the reference list, consequently making it  difficult to find the quoted publication.

Author Response

(The authors gave the same response as above.)

Reviewer 3 Report

As far as I know, there have been many previous studies on organic matter transformation in organic soils, but further studies are still required for understanding in more detail. Therefore, I think the aim of this research is important in this field of research. In general, I am afraid that discussion on the results of the experiments is not sufficient. The main conclusion of the study is not clear and persuasive. In addition, the manuscript has not been well prepared yet for submission; it still includes many errors in citations and methodology. Many statements are unclear, even in the abstract. Therefore, I invite you to resubmit after careful revision. 

I hope the following comments will be helpful for revising the manuscript.

1.     The title is not appropriate. I suggest to change the title of the paper by indicating some novel results/methodology and significance.

2.     The importance of the study should be describe in the introduction with recent citations. 

3.     There is problem with transmission of sentences. No links have been developed. 

4.     The sentence is lack of understanding. Please rephrase the sentence to convey your message. 

5.     A lot of work has been done on soil organic matter transformation. How is your work different and needed?

6.     Abstract looks very general. Numerical results should be included in the abstract.

7.     While using abbreviation, use full of abbreviation at first. i.e., L25, DON

8.     Better to add p-values while indicating significant results. 

Author Response

Dear Reviewer,

Thank you for your review. Your suggestions have been taken into account, namely:

  • we changed the title of our manuscript,
  • the importance of our research was described in more detail in the introduction,
  • some sentences have been reformatted,
  • we wrote a new summary,
  • we changed the symbols of obtained nitrogen fractions,
  • we justified the need for our research,
  • we extended the discussion and supplemented the references with 13 references.

Sincerely

Authors

Author Response

(The authors gave the same response as above.)

Round 2

Reviewer 2 Report

Reference Lines 547;561-562; 563; 602; 604; 608; 616: References  not prepared in accordance with the requirements journal  IJERPH

Author 1, A.B.; Author 2, C.D. Title of the article. Abbreviated Journal Name Year, Volume, page range.

After minor technical corrections, the manuscript is suitable for publication.

Reviewer 3 Report

I recommend this article for publication